# Assessing and Improving the Accuracy of Visible Infrared Imaging Radiometer Suite Ocean Color Products in Environments with High Solar Zenith Angles

**Hao Li** [1], **Xianqiang He** [2,3,*], **Palanisamy Shanmugam** [4], **Yan Bai** [2,3], **Difeng Wang** [2], **Teng Li** [2] and **Fang Gong** [2]

1. Donghai Laboratory, Zhoushan 316021, China; lihao@sio.org.cn
2. State Key Laboratory of Satellite Ocean Environment Dynamics, Second Institute of Oceanography, Ministry of Natural Resources, Hangzhou 310012, China; baiyan@sio.org.cn (Y.B.); dfwang@sio.org.cn (D.W.); liteng@sio.org.cn (T.L.); gongfang@sio.org.cn (F.G.)
3. School of Oceanography, Shanghai Jiao Tong University, Shanghai 200030, China
4. Ocean Optics and Imaging Laboratory, Department of Ocean Engineering, Indian Institute of Technology Madras, Chennai 600036, India; pshanmugam@iitm.ac.in
* Correspondence: hexianqiang@sio.org.cn

**Abstract:** Utilizing in situ measurement data to assess satellite-derived long-term ocean color products under different observational conditions is crucial for ensuring data quality and integrity. In this study, we conducted an extensive evaluation and analysis of Visible Infrared Imaging Radiometer Suite (VIIRS) remote sensing reflectance (Rrs) products using long-term OC-CCI in situ data from 2012 to 2021. Our research findings indicate that, well beyond its designed operational lifespan, the root mean square difference accuracy of VIIRS Rrs products across most spectral bands remains superior to 0.002 (sr$^{-1}$). However, VIIRS Rrs products in shorter wavelength bands (e.g., at 412 nm) have exhibited significantly lower accuracy and a long-term bias in recent years. The annual precision of VIIRS Rrs products demonstrated a declining trend, particularly in coastal or eutrophic waters. This degradation in accuracy highlights the imperative for continuous monitoring of VIIRS performance and further advancements in the atmospheric correction algorithm, especially to address satellite records at high solar zenith angles (SZAs) and observation zenith angles (OZAs). Our analysis indicates that, in observation environments with high SZAs (greater than 70°), the accuracy of VIIRS Rrs products has declined by nearly 50% compared to typical solar zenith angle observation conditions. To address the challenge of declining accuracy under large observation geometries, we introduced the neural network atmospheric correction model (NN-V). Developed based on meticulously curated VIIRS products, the NN-V model exhibits outstanding performance in handling VIIRS data in conditions of extensive observation geometries. During the winter season in high-latitude marine regions, the NN-V model demonstrates a remarkable enhancement in ocean color product coverage, achieving an increase of nearly 20 times compared to traditional methods.

**Keywords:** ocean color; VIIRS; atmospheric correction; diurnal change; large solar zenith angle

## 1. Introduction

The Visible Infrared Imaging Radiometer Suite (VIIRS) instrument is the successor to the Moderate Resolution Imaging Spectroradiometer (MODIS) for generating ocean color and earth data products. The VIIRS is a multidisciplinary instrument mounted on the Joint Polar Satellite System (JPSS)—Suomi National Polar-orbiting Partnership Satellite (S-NPP), launched in October 2011. The VIIRS sensor features 22 spectral bands ranging from 412 nm to 12 μm, including 16 moderate-resolution bands with a maximum resolution of 750 m, 5 high-resolution bands at 375 m, and a day/night band (DNB) suitable for global observations of the land, oceans, atmosphere, and cryosphere. The VIIRS data products have been made available from 2012 to 2023, spanning approximately 12 years,

which notably exceeds the planned life expectancy of 5 to 7 years. Recent studies have indicated that the MODIS/Aqua longevity is impressive, as it continued to operate over a period of 21 years beyond its design lifetime of 6 years, despite minor adjustments made to further improve the retrieved ocean color products. The long-term ocean color records of the MODIS/Aqua instrument also led to notable performance degradation and indicated the need for new calibration methods. For instance, Meister et al. (2014) proposed a calibration method specifically targeting the MODIS bands. This method only utilized the central part of the MODIS scan to generate the averaged L3 data products at the desired spatial and temporal scales [1]. Using the restricted L3 data accounting for scan angles and cross-calibration coefficients for each scan angle, the cross-calibration coefficients for the central part were determined to be close to 1 as an indicator of good calibration. The residual trends persisted on the scan edges, particularly at the 412 and 443 nm bands. For the 412 nm band, the correction at the scan edges reached up to 3%. When evaluating the long-term VIIRS ocean color products, Cao et al. (2013) found that the degradation of the mirror component of the VIIRS rotating telescope stabilized at approximately 30% [2]. However, continuous monitoring of its performance over a long period is necessary. Uprety et al. (2015) conducted an analysis of the radiometric performance of the VIIRS sensor by comparing it with other satellite instruments such as MODIS/Aqua and Landsat 8 OLI [3]. The results showed that the stability of most VIIRS bands was better than 0.5% and the uncertainty was around 1%. After accounting for spectral differences, the absolute radiance deviation estimated through cross-calibration between VIIRS and MODIS was less than 2% for each waveband. In general, similar to the MODIS/Aqua instrument calibration activities, it is critical to assess the long-term spatial and temporal ocean color products of VIIRS and ensure their accuracy and integrity for global ocean applications on a regular basis.

Most of the existing validations of VIIRS products have primarily relied on data collected from fixed platforms (such as buoys) or AERONET-OC observation stations installed on offshore oil platforms. During the early launch phase of VIIRS, Hlaing (2013) validated the accuracy of VIIRS products based on nearly one year of data (2012) from two coastal AERONET-OC (LISCO and WaveCIS) sites. The results showed that the VIIRS products were capable of capturing seasonal and temporal variations of water properties, with an average correlation coefficient greater than 0.96 for all bands except the 412 nm band and an average absolute percentage difference (APD) of approximately 20% [4]. However, the VIIRS products were underestimated at 412 nm bands. To address this issue, Hlaing et al. (2014) performed extensive radiative transfer simulations of the coupled ocean–atmosphere system using the aerosol optical properties and outgoing radiance data from the AERONET-OC sites and performed the radiometric proxy calibration of VIIRS in visible and near-infrared bands. The difference in the blue band was approximately 0.5% [5]. Wang et al. (2015) evaluated the VIIRS ocean color products using data from the Marine Optical Buoy (MOBY) near Lanai Island, Hawaii, which included the normalized water-leaving radiance spectra (nLw($\lambda$)) and chlorophyll-a (Chl-a) concentration in five bands of VIIRS [6]. The results showed that the VIIRS Chl-a concentrations in global oligotrophic waters were significantly lower in 2013 than in 2012, while exhibiting little inter-annual variation in MODIS/Aqua products between 2012 and 2013. This indicates a serious issue with the VIIRS calibration in the visible bands. It was primarily related to the attenuation values at the VIIRS 551 nm band. Vandermeulen et al. (2015) validated the accuracy of VIIRS products using in situ data from the AERONET-OC sites located in the Chesapeake Bay and Mississippi River plume, and the results showed a root mean square difference (RMSD) value of 0.160 mW cm$^{-2}$m$^{-1}$sr$^{-1}$ for the 551 nm band [7]. Brando et al. (2016) reported spectral comparisons of radiometer data with VIIRS in different tropical water types off northern Australia based on the standard NIR atmospheric correction algorithm (available in SeaDAS 8.1 software) [8]. The results showed a high consistency (RMSD < 0.002 sr$^{-1}$) for all wavelengths above 530 nm, but the satellite reflectance data consistently underestimated the in situ spectra in blue bands by 7.5–29%. After several

years of VIIRS operation, Barnes et al. (2019) validated the VIIRS remote sensing reflectance products using in situ measurements from 53 cruises in nearshore regions of the Gulf of Mexico between 2012 and 2017. The results revealed APD values of 37%, 30%, 22%, 23%, and 43% for the 412 nm, 443 nm, 486 nm, 551 nm, and 671 nm bands, respectively [9].

Using data from fixed platforms is of great advantage because of their high temporal resolution. The use of high-frequency observations from fixed platforms with small time intervals (within 1 h) also leads to more accurate validation of the satellite-derived ocean color products. However, certain distinctly different areas of algal blooms and river plumes are often under-sampled by these methods. Most prior studies have primarily focused on validating satellite data products for specific years without considering the long-term quality and integrity of the products. Moreover, there has been a lack of precision testing for VIIRS products under extreme observational geometries, such as high solar zenith angle (SZA) or observation zenith angle (OZA) conditions. Barnes and Hu's (2016) study delves into the angular dependence in single-sensor measurements and cross-sensor consistency, highlighting that VIIRS remote sensing reflectance (Rrs) products experience substantial impact when the OZA surpasses 40° [10]. Therefore, the main objective of this study is to evaluate the long-term accuracy of VIIRS Rrs products, specifically under high SZA or OZA conditions. To achieve this goal, a comprehensive dataset covering open ocean waters, estuaries, river plumes, algal blooms, and coastal areas was utilized for assessment, with in situ data collected from fixed observation platforms and ship measurements. This holistic approach aims to provide insights into the extended accuracy and precision of VIIRS products over an extended period and in challenging observational scenarios. Furthermore, new models were developed to enhance the accuracy of VIIRS ocean color products. The incorporation of a neural network model was explored to establish an atmospheric correction model tailored for VIIRS data under high SZA observation conditions. This effort is crucial for improving the precision of VIIRS products in such observation environments. By addressing the limitations of previous research and incorporating innovative modeling techniques, this study contributes to a more robust understanding of the long-term performance and accuracy of VIIRS Rrs products, particularly in the context of high SZA observations.

## 2. Data and Methods

### 2.1. In Situ Data

The in situ measurements of remote sensing reflectance (Rrs) are the primary data used to evaluate the VIIRS products' accuracy. These data were obtained from a database created by Valente et al. who previously utilized this database to validate the OC-CCI (ESA Ocean Color Climate Change Initiative) ocean color products [11–13]. It includes a vast number of measurements made during the period from 1997 to 2021, comprising six commonly used data sources for ocean color validation (MOBY, BOUSSOLE, AERONET-OC, SeaBASS, NOMAD, and MERMAID) and four data sources for ocean color applications (AWI, COASTCOLOUR, TPSS, and TARA). Any duplicate data within these data sources were screened out; thus, priority was given to the NOMAD dataset, followed by the data from the individual projects (MOBY, BOUSSOLE, AERONET-OC, AMT, HOT, and GeP&CO) and other sources (SeaBASS, MERMAID, and ICES). This approach was chosen due to its global popularity and wider utility in ocean color work. The compiled in situ Rrs spectra were normalized to a single sun-viewing geometry (sun at zenith and nadir viewing) and corrected for the bidirectional effects (Morel et al., 2002) [14]. Furthermore, Valente et al. applied homogenization, quality control, and merging methods to all the data. Minimal changes were made to the original data, other than averaging the observations that were closely collected in time and space, eliminating some points after quality control, and converting these data to a standard format [11–13].

During the operational time period of VIIRS, we obtained the remote sensing reflectance from OC-CCI in situ data covering the period from 1 January 2012 to 27 September 2021 (insitudb_rrs_satbands6_V3, https://doi.org/10.1594/PANGAEA.941318, ac-

cessed on 1 October 2023). This dataset includes approximately 32,198 in situ measurements of Rrs spectra (values are generally available at 412 nm, 443 nm, 486 nm, 551 nm, and 671 nm), of which approximately 30,000 measurements were collected on fixed platforms such as AERONET-OC and BOUSSOLE and the remaining 2000 measurements on mobile platforms like NOMAD and others. Due to clouds, sun glints, straylight, and other factors, the number of matchups between in situ and satellite Rrs data was significantly lower than the total number of measurements. The distribution of the sampling locations is shown in Figure 1, where much of these data come from Case 2 waters and a small portion from open oceanic waters (Case 1).

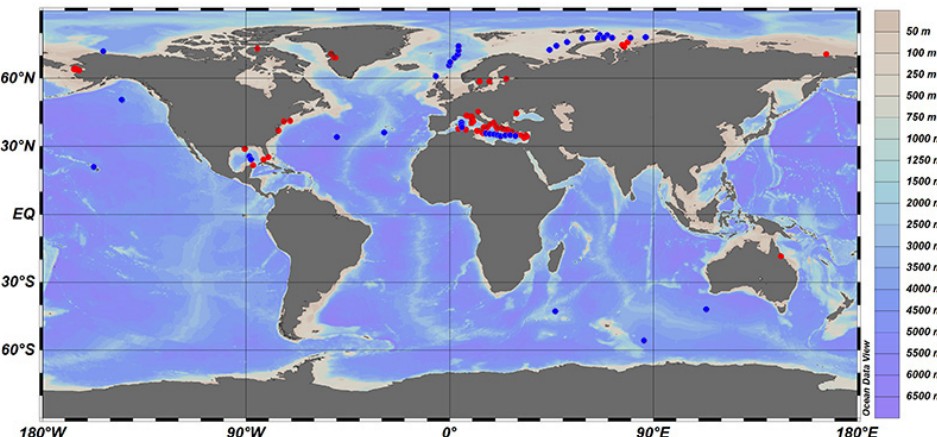

**Figure 1.** Distribution of sampling locations of the OC-CCI Rrs measurements from 2012 to 2021 (red dots represent Case 2 waters, and blue dots represent Case 1 waters).

### 2.2. Satellite Data

The VIIRS Level 2 products were generated by the NASA OBPG (http://oceancolor. gsfc.nasa.gov, accessed on 2 October 2023) using SeaDAS 8.1 software (with the standard iterative NIR atmospheric correction algorithm) and were obtained from the sampling locations of OC-CCI in situ data. These data were processed using the latest processing scheme (version 2022.0), which includes updates on VIIRS instrument calibration for the radiometric degradation issues based on the on-board (solar/lunar) measurements. The wavelengths of the VIIRS sensor slightly differed from those of the OC-CCI in situ dataset, and hence all comparisons were made using the reference VIIRS wavelengths (412, 443, 486, 551, and 671 nm).

For the matchup comparison, the data processing excluded the individual pixels meeting any of the following conditions: land, cloud, failure in atmospheric correction, stray light, bad navigation quality, high and moderate glint, and negative Rayleigh-corrected radiance. In addition, pixels with negative values at any of the wavelengths of the water-leaving radiance spectra were also excluded from spatial averaging. A SZA threshold was fixed to 90 degrees to evaluate the accuracy of satellite products under different SZAs and to obtain the corresponding products.

### 2.3. K-Means Clustering Analysis

Due to the extensive volume of in situ data used, we conducted K-Means clustering analysis on all the employed in situ data to enhance the clarity of the dataset structure. K-Means clustering analysis is a commonly used unsupervised learning algorithm designed to partition observation points in a dataset into different groups, ensuring that points within the same group are similar while those in different groups exhibit significant differences. K-Means clustering performs well when the structure of the dataset is evident [15]. The algorithm involves the following steps: 1. Randomly select K initial centroids, which represent the centers of K groups in the dataset. 2. Allocate each observation point to the group represented by the nearest centroid. 3. Calculate the new centroid for each group,

representing the average of all observation points in that group. 4. Repeat the assignment and update steps until the centroids no longer undergo significant changes or the specified number of iterations is reached. We utilized an empirical approach to establish the optimal number of clusters for K-Means clustering. Initially, we explored a broader range of cluster numbers and refined the final count by closely examining the clustering results. This adjustment was guided by an analysis of the rate of decrease in the within-cluster sum of squares across different cluster numbers. Our decision-making process was informed by the actual performance of the data, ensuring that the determined number of clusters was well-suited for our dataset.

### 2.4. Neural Network Atmospheric Correction Model

In addressing the challenge of diminished accuracy in VIIRS products under extensive observational geometric conditions, this study employs a neural network atmospheric correction model to process VIIRS Level 1 data [16,17]. The utilized neural network atmospheric correction model is built upon the framework established by Li et al., which was originally designed for processing GOCI (Geostationary Ocean Color Imager) and MODIS (Moderate Resolution Imaging Spectroradiometer) satellite data.

The neural network training dataset for this model is curated from VIIRS Level 1 data spanning the years 2016 to 2017. To ensure the representativeness of the established training dataset across diverse seasons, multiple satellite images with a cloud cover of less than 60% are meticulously chosen for each month. The data undergo filtering based on criteria that include parameters such as the mean, variance, and coefficient of variation, ensuring the high quality of the selected remote sensing reflectance products. This filtering process guarantees consistency in spatial and temporal dimensions, and detailed information can be referenced in the literature by Li et al. (2020) [18].

Following the extraction of a high-quality dataset featuring remote sensing reflectance with small SZAs, this dataset is utilized to correlate remote sensing reflectance with larger SZAs and Rayleigh-corrected radiance at the same location and time window, thereby constituting the training dataset. Then, the full training dataset was separated into a model training dataset and a model testing dataset based on a randomly selection with percentages of 70% and 30%, respectively. Upon inputting the model training dataset into the neural network, the study establishes a tailored neural network atmospheric correction model (NN-V) designed specifically for VIIRS data under extensive observational geometric conditions. The constructed neural network model consists of a single layer, comprising an input layer, an intermediate layer, and an output layer. The input data encompass Rayleigh-corrected radiance in various bands of VIIRS visible light, SZA, satellite zenith angles, and relative azimuth angles. The output layer provides remote sensing reflectance for each band of VIIRS visible light.

### 2.5. Temporal and Spatial Matching Scheme and Statistical Parameters

To validate the satellite products, we considered the significant differences/variations in the time matching window, pixel box size, and coefficient of variation (CV). For example, the time matching window can be $\pm3$ h or $\pm5$ h, the pixel box can be $3 \times 3$ or $5 \times 5$, and the coefficient of variation can be 0.15, 0.2, and 0.4. Studies by Mélin et al. (2007) and Barnes et al. (2015) have indicated that the statistical errors remain relatively consistent across different time matching windows and coefficients of variation [10,19]. In this study, the spatial and temporal matching of in situ and satellite data was successfully done by averaging all pixels in the $3 \times 3$ window centered on the observation point with a time window of $\pm3$ h [20,21]. Additionally, the following conditions were applied:

(1) The percentage of valid pixels in the $3 \times 3$ window was checked. If it exceeded 50%, the data were used and otherwise discarded.

(2) The mean and standard deviation (SD) of all validation pixels were calculated. Any pixel falling outside the range of the mean $\pm 1.5$ SD was removed.

(3) For the remaining pixels, the CV, calculated as SD/mean, was used to ensure spatial consistency. If the CV exceeded 0.15, the data were discarded.

For pixels meeting the aforementioned criteria, we conducted a comparison analysis between the in situ and satellite-derived remote sensing reflectance products. This approach excluded the data affected by unexpected changes in natural and environmental conditions as well as artifacts in the satellite-derived products resulting from the critical sensor characteristics using the filtering procedures as previously outlined in this section.

Statistical evaluation of the VIIRS Rrs product accuracy is based on global metrics such as the square of the Pearson product–moment correlation coefficient ($R^2$), adjusted $R^2$ ($R^2_{adj}$), APD, relative percentage difference (RPD), and RMSD. These metrics are defined below [22,23],

$$R^2 = \frac{\left( \sum (x_i - \bar{x})(y_i - \bar{y}) \right)^2}{\sum (x_i - \bar{x})^2 \sum (y_i - \bar{y})^2} \tag{1}$$

$$R^2_{adj} = 1 - \frac{(1 - R^2) * (N - 1)}{(N - 2)} \tag{2}$$

$$\text{RMSD} = \sqrt{\frac{\sum_{i=1}^{N}(y_i - x_i)^2}{N}} \tag{3}$$

$$\text{APD} = 100\% * \frac{1}{N}\sum_{i=1}^{N}\frac{|y_i - x_i|}{x_i} \tag{4}$$

$$\text{RPD} = 100\% * \frac{1}{N}\sum_{i=1}^{N}\frac{y_i - x_i}{x_i} \tag{5}$$

where $x_i$, $y_i$, and $N$ represent the in situ values, retrieved Level 2 products, and the number of samples, respectively. The adjusted $R^2_{adj}$ is proficient in presenting $R^2$ values that have been corrected to mitigate inflation caused by variations in sample size. APD quantifies the systematic error. RPD is a primary metric used to assess the accuracy and bias in satellite-derived products. RMSD takes into account the mean and variance of the error distribution to define the random error. Using these multiple statistical metrics is an effective approach to evaluating the satellite products' accuracy.

## 3. Results

This section presents the in situ spectral characteristics of different waters, overall accuracy and its variation with observation geometry, impacts of observation geometry, and long-term accuracy variation in the water color products.

### 3.1. In Situ Spectral Characteristics

A total of 8312 satellite spectral data were matched with in situ data. Figure 2 displays the spectral plots of the results from the clustering analysis based on the OC-CCI in situ dataset. The data can be broadly divided into four categories: (a) turbid continental shelf waters, (b) clear continental shelf waters, (c) turbid coastal waters, and (d) clear open ocean waters. These four water types account for 24%, 54%, 12%, and 10%, respectively. The overall range of remote sensing reflectance at 551 nm for these water types is from 0.0009 to 0.027 sr$^{-1}$.

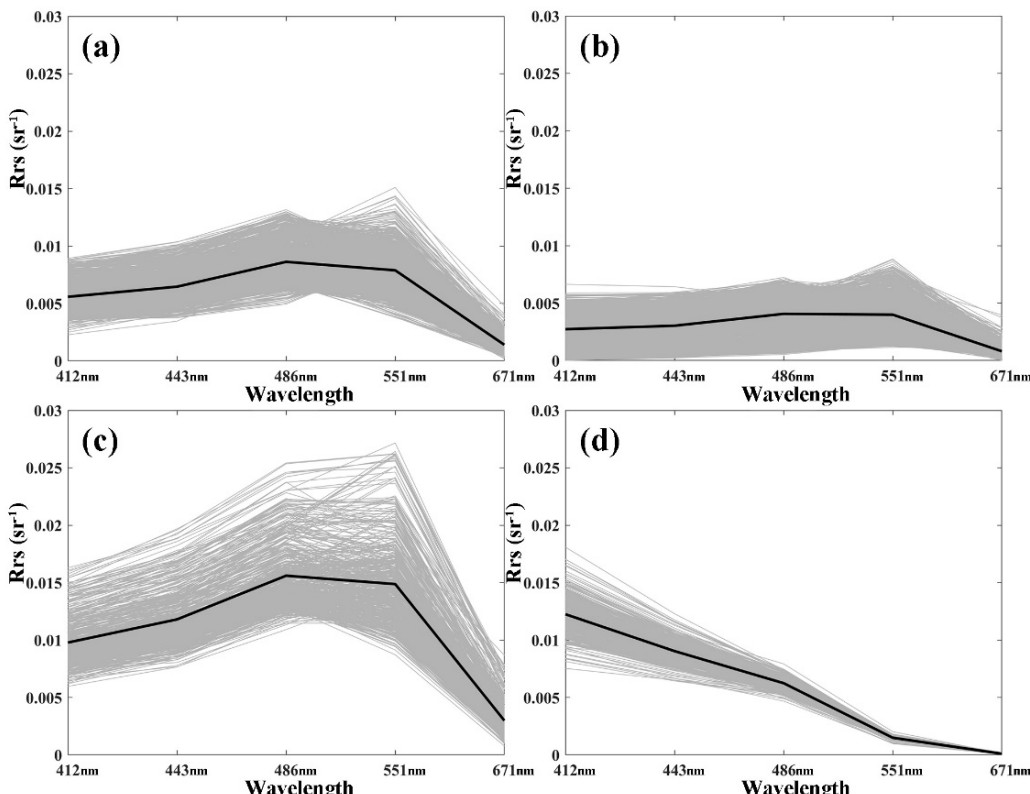

**Figure 2.** Spectral plots of the results from the clustering analysis based on the in situ data. The heavy black line represents the mean value. (**a**) turbid continental shelf waters, (**b**) clear continental shelf waters, (**c**) turbid coastal waters, and (**d**) clear open ocean waters.

*3.2. Overall Assessments*

Figure 3 presents the VIIRS remote sensing reflectance (Rrs) products for each band versus the OC-CCI in situ data. The scatter points for each band are closely aligned with the 1:1 line, demonstrating a good correlation between the retrieved and measured Rrs data over a period of 13 years and indicating high accuracy of the VIIRS Rrs products in various water types. Table 1 summarizes the overall statistical results. The accuracy of the Rrs at 486 nm is generally higher compared to other bands, with an APD of 17.9% and an RPD of −2.45%. Across all VIIRS bands, there is a noticeable underestimation of Rrs values, as indicated by the negative RPD values.

**Table 1.** Statistical results for the VIIRS remote sensing reflectance products versus the OC-CCI validation data.

| Bands | N | $R^2$ | RMSD ($sr^{-1}$) | APD (%) | RPD (%) | Slope |
|---|---|---|---|---|---|---|
| 412 nm | 8312 | 0.680 | 0.0019 | 46.4 | −15.00 | 0.87 |
| 443 nm | 8312 | 0.695 | 0.0018 | 30.4 | −15.41 | 0.94 |
| 486 nm | 8312 | 0.760 | 0.0018 | 17.9 | −2.45 | 0.97 |
| 551 nm | 8312 | 0.660 | 0.0031 | 26.3 | −13.60 | 1.02 |
| 671 nm | 8312 | 0.501 | 0.0010 | 49.7 | −18.24 | 1.14 |

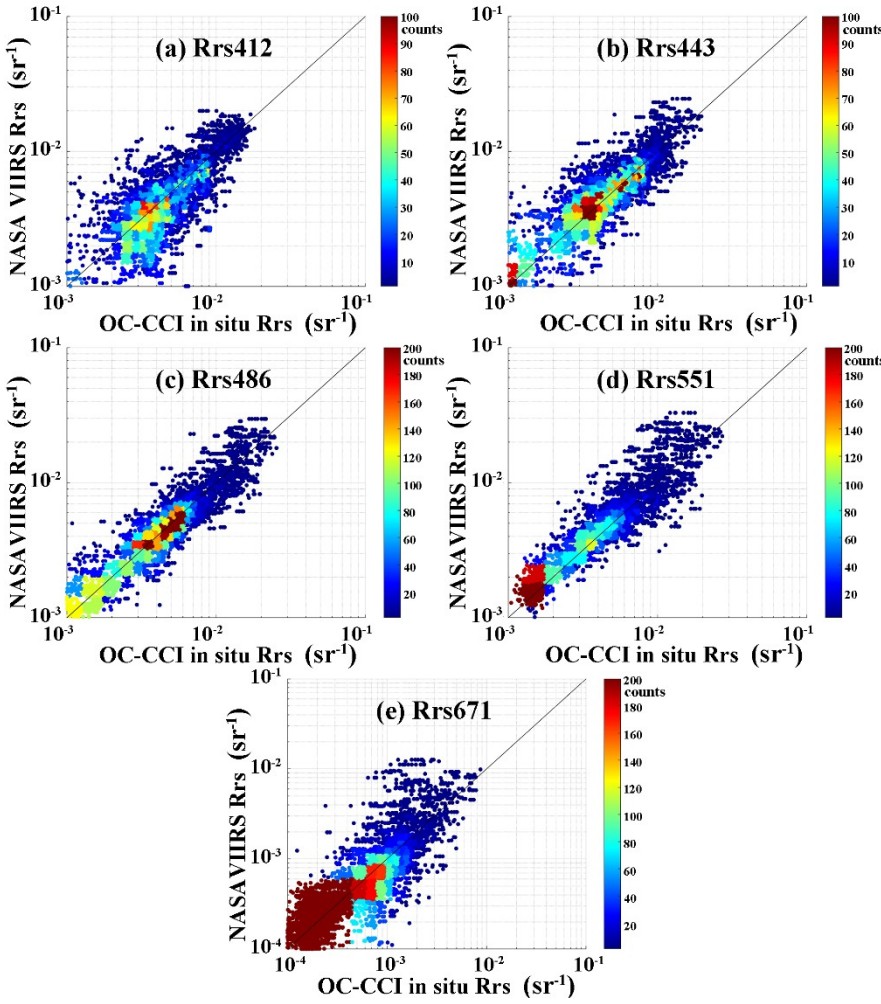

**Figure 3.** Evaluation of the VIIRS Rrs products for each band using OC-CCI in situ data (the black line in the graph represents the 1:1 line).

### 3.3. Annual Variation in Product Accuracy

After assessing the overall accuracy of VIIRS Rrs products, we further investigated the accuracy of annual products at 486 nm from 2012 to 2021 (Figures 4 and 5). Notably, when the Rrs values at 486 nm exceed 0.01 $sr^{-1}$, the validation data points are significantly scattered and deviate from in situ measured data. This indicates that the inversion accuracy of VIIRS Rrs products is higher in clear water bodies (typically with Rrs(486) less than 0.01 $sr^{-1}$) compared to coastal waters (typically with Rrs(486) greater than 0.01 $sr^{-1}$). Moreover, we observed a declining trend in the accuracy of VIIRS Rrs products over the years. The accuracy remains stable in clear waters but deteriorates in water bodies with Rrs(486) greater than 0.01 $sr^{-1}$ (typically coastal turbid and eutrophic waters), as evidenced by the wide scatter points and increased deviation in the annual products. Figure 5 illustrates the temporal variation of the APD for Rrs(486) nm. The red line in the figure represents the linear regression line with a slope of 0.005, indicating an increase above zero. This also suggests a decline in product accuracy over time. Table 2 presents the statistical parameters of the accuracy assessment of VIIRS-Rrs(486) products on a yearly basis. Although the number of measured data used for this evaluation impacts the final statistical parameters, the measured data exceeded 300 for all years except 2021. Using similar amounts of measured data (such as 2014 (465 data points) and 2020 (439 data points)), we found that the $R^2_{adj}$ and APD values were 0.744 and 16.46% and 0.607 and 20.03%, respectively. Compared to 2014, there has been a significant 21.6% decrease in the APD and an 18.4% decrease in $R^2$ in 2020. Upon calculation, the annual change in slope for

$R^2$ is $-0.02$, and for APD, it is 0.38. These figures point to a noticeable decline in product accuracy. Additionally, Table 2 presents the APD for Rrs(486) when greater than or less than 0.01 sr$^{-1}$, suggesting that the decline in product accuracy is primarily attributed to turbid water bodies.

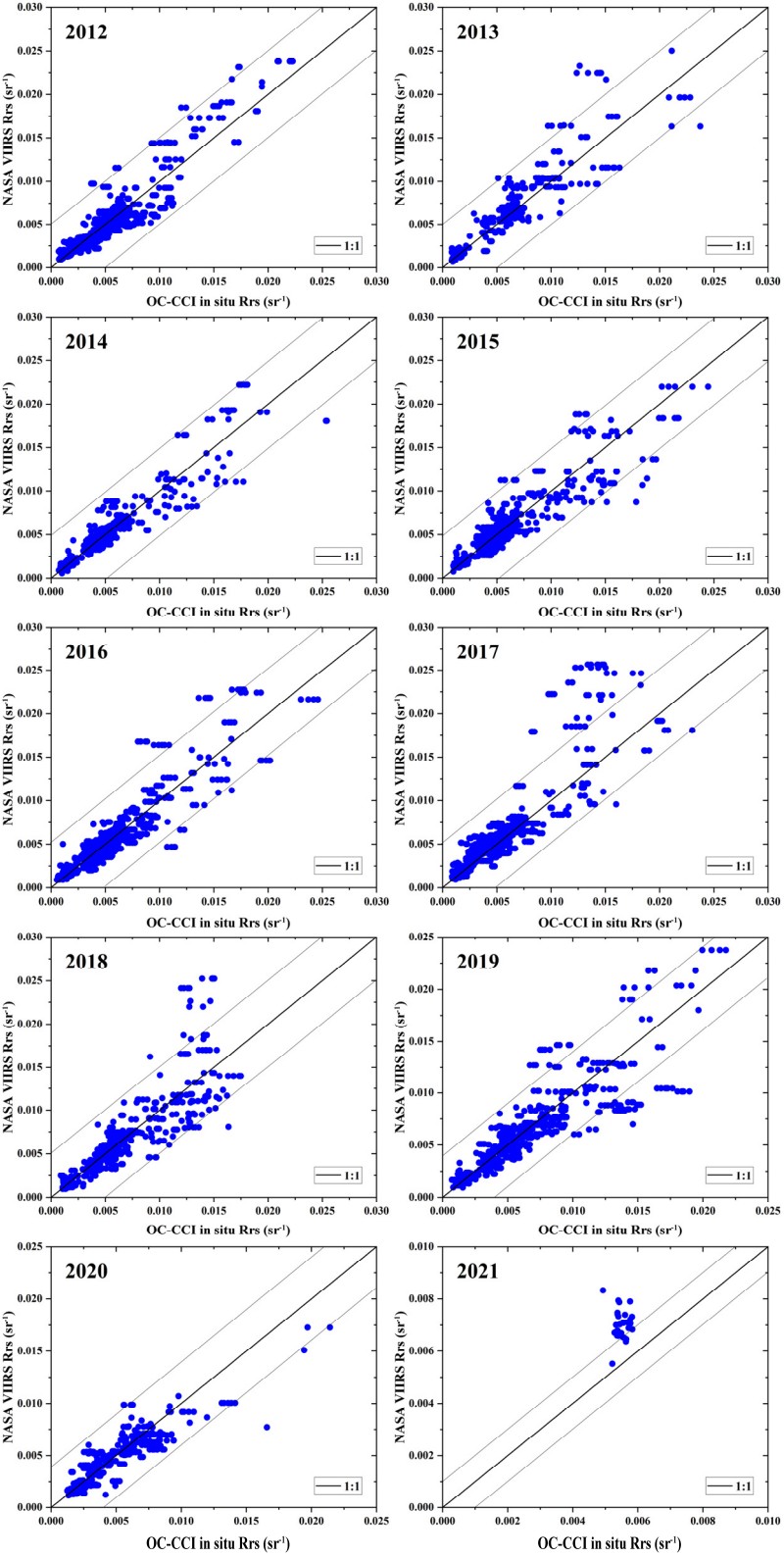

**Figure 4.** Evaluation of the VIIRS Rrs(486) products using OC-CCI in situ data in different years (2012~2021).

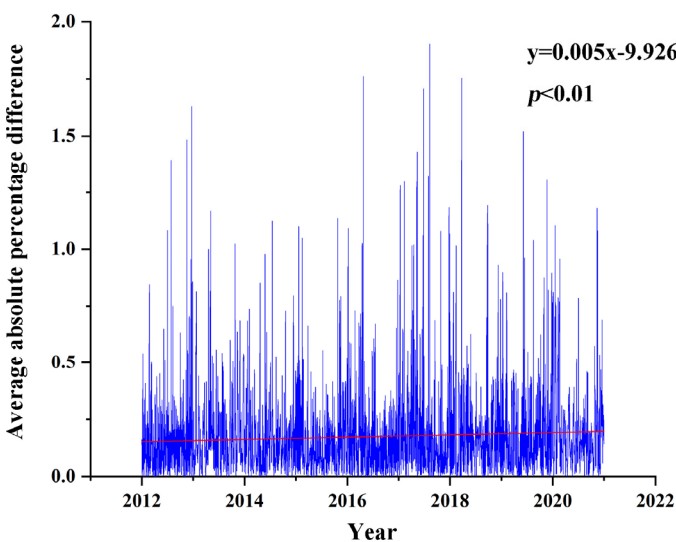

**Figure 5.** Temporal variation of APD for Rrs(486)nm (2012~2021). The red line represents the linear regression fit.

**Table 2.** Statistical results for the VIIRS Rrs(486) products of 486 nm versus the OC-CCI in situ data in different years (2012~2020).

| Years | N | $R^2_{adj}$ | RMSD (sr$^{-1}$) | APD (%) | RPD (%) | Rrs < 0.01 sr$^{-1}$ APD (%) | Rrs > 0.01 sr$^{-1}$ APD (%) |
|---|---|---|---|---|---|---|---|
| 2012 | 1388 | 0.840 | 0.0013 | 16.43 | 3.25 | 16.09 | 21.10 |
| 2013 | 354 | 0.822 | 0.0021 | 20.01 | −6.20 | 18.48 | 24.35 |
| 2014 | 465 | 0.863 | 0.0015 | 16.46 | −2.26 | 16.39 | 17.56 |
| 2015 | 1015 | 0.786 | 0.0014 | 15.52 | −0.32 | 14.65 | 20.51 |
| 2016 | 1650 | 0.826 | 0.0013 | 14.49 | 0.49 | 13.85 | 21.22 |
| 2017 | 1472 | 0.730 | 0.0020 | 20.67 | −10.36 | 18.89 | 49.41 |
| 2018 | 610 | 0.748 | 0.0022 | 19.50 | 0.70 | 18.24 | 26.76 |
| 2019 | 889 | 0.746 | 0.0019 | 19.07 | −3.02 | 18.13 | 26.83 |
| 2020 | 439 | 0.779 | 0.0014 | 20.03 | −1.80 | 19.49 | 23.04 |

### 3.4. Variation of Product Accuracy with Observation Geometry

The accuracy of VIIRS Rrs products was assessed for different SZAs and OZAs. To better utilize statistical metrics to evaluate VIIRS Rrs products at different SZAs, we standardized the data. For instance, in the SZA range from 40° to 50°, the actual matched data size was 1644, but we randomly selected 790 data points to match the minimum available data size in the SZA range greater than 70°. Figure 6 presents the evaluation results, where, except for the 551 nm band, data points are notably concentrated on the 1:1 line under typical SZAs (30° to 60°), while under higher SZAs (>70°), data points exhibit a distinct scattering pattern across all bands. Table 3's statistical metrics clearly indicate that under normal/smaller SZAs, there is no clear trend in product accuracy with changing SZAs, and errors and deviations are small. However, there is a significant decline in product accuracy, which is consistent with the findings of previous studies [24,25]. Figure 7 and Table 4 present the evaluation results for VIIRS Rrs products under different OZAs. Similar to SZAs, there is no clear trend in product accuracy with changing OZAs when the OZA is less than 60°. However, at OZAs greater than 60°, data points are more dispersed, and APD is significantly larger, indicating a noticeable decline in product accuracy.

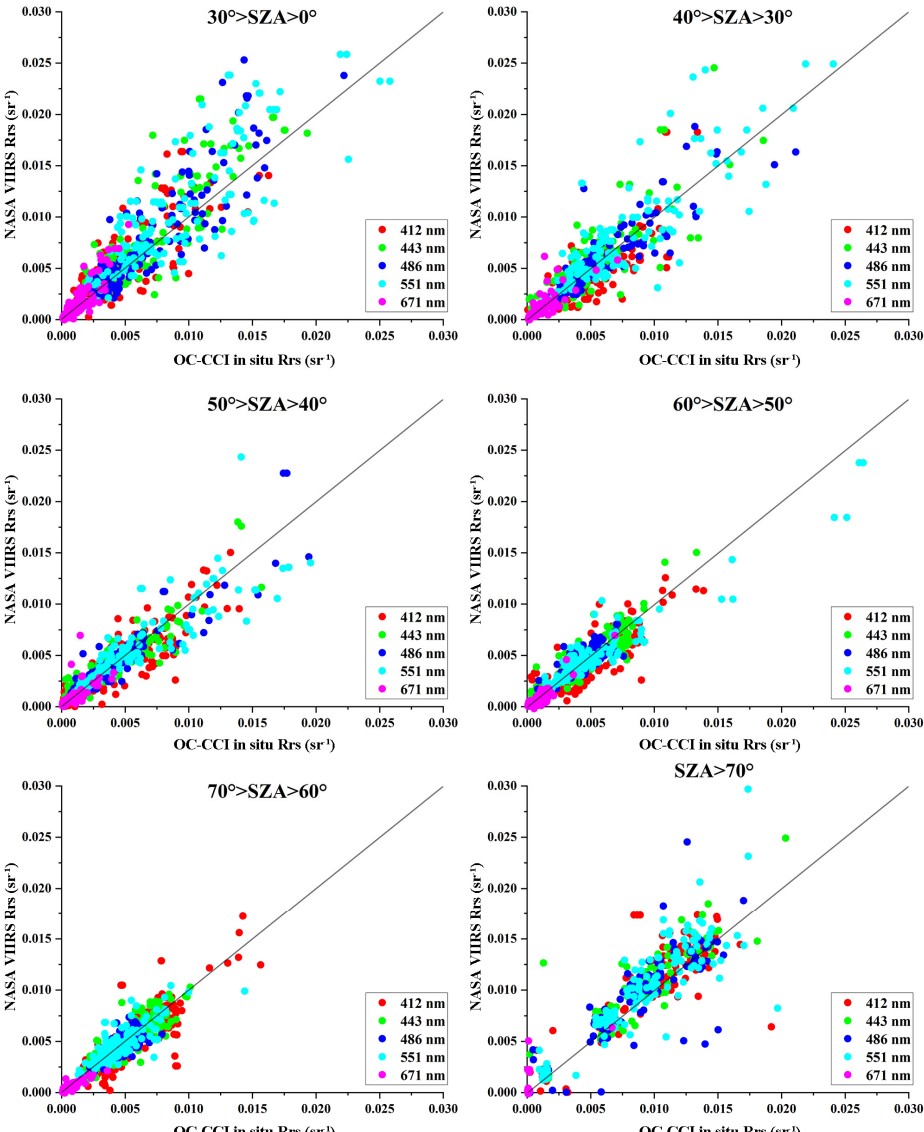

**Figure 6.** Evaluation of the VIIRS Rrs products using OC-CCI in situ data under different SZAs (the black line in the graph represents the 1:1 line).

**Table 3.** Statistical results for the VIIRS Rrs products versus OC-CCI in situ Rrs data under different SZAs.

| SZA | N | $R^2$ | RMSD $(sr^{-1})$ | APD (%) | RPD (%) | Slope |
|---|---|---|---|---|---|---|
| $30° > SZA > 0°$ | 790 | 0.795 | 0.0026 | 34.96 | −17.51 | 0.86 |
| $40° > SZA > 30°$ | 790 | 0.771 | 0.0019 | 33.84 | −12.39 | 0.92 |
| $50° > SZA > 40°$ | 790 | 0.836 | 0.0012 | 29.27 | −6.26 | 0.97 |
| $60° > SZA > 50°$ | 790 | 0.873 | 0.0012 | 23.07 | −1.67 | 0.95 |
| $70° > SZA > 60°$ | 790 | 0.830 | 0.0014 | 33.99 | −14.07 | 0.84 |
| $SZA > 70°$ | 790 | 0.876 | 0.0020 | 48.69 | −38.59 | 0.79 |

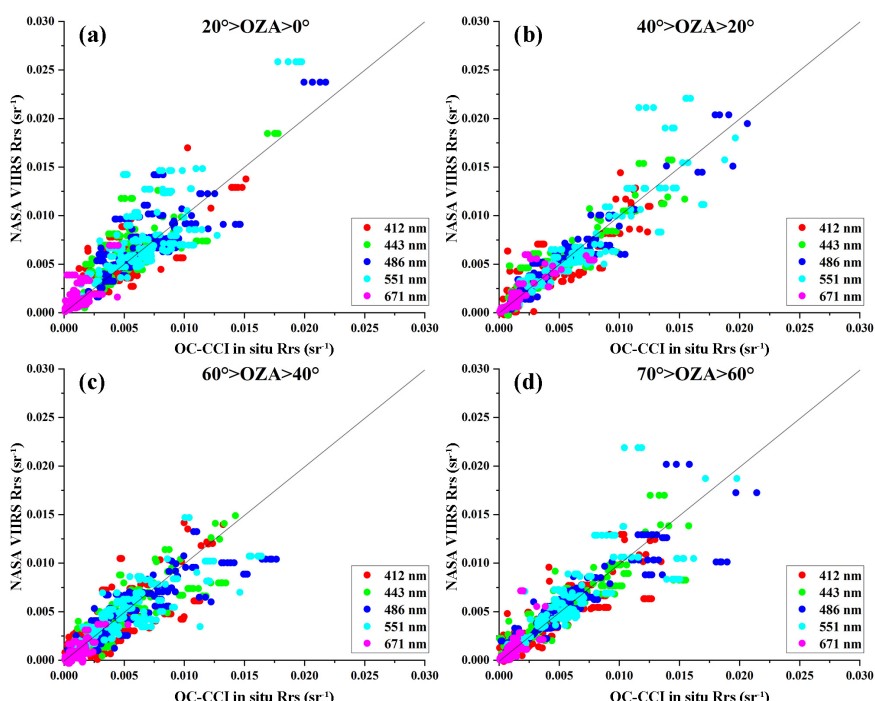

**Figure 7.** Evaluation of the VIIRS Rrs products using OC-CCI in situ data under different OZAs. (The black line in the graph represents the 1:1 line). (**a**) OZA ranges from 0° to 20°. (**b**) OZA ranges from 20° to 40°. (**c**) OZA ranges from 40° to 60°. (**d**) OZA ranges from 60° to 70°.

**Table 4.** Statistical results of the VIIRS Rrs products versus OC-CCI in situ Rrs data under different OZAs.

| OZA | N | $R^2$ | RMSD ($sr^{-1}$) | APD (%) | RPD (%) | Slope |
|---|---|---|---|---|---|---|
| 20° > OZA > 0° | 970 | 0.732 | 0.0019 | 38.48 | −19.45 | 0.96 |
| 40° > OZA > 20° | 970 | 0.843 | 0.0015 | 34.18 | −5.39 | 0.93 |
| 60° > OZA > 40° | 970 | 0.717 | 0.0018 | 38.85 | −3.75 | 0.82 |
| 70° > OZA > 60° | 970 | 0.790 | 0.0018 | 43.42 | −21.29 | 0.74 |

*3.5. Enhanced VIIRS Product Accuracy through Neural Network Atmospheric Correction Model*

Based on the VIIRS dataset generated in Section 2.3, a neural network model, NN-V, tailored for VIIRS products, was developed. The training efficacy of the NN-V model was initially assessed using a model testing dataset. The results are illustrated in Figure 8. It is evident that the NN-V model performs well in the retrieval of various VIIRS bands, with calculated $R^2$ exceeding 0.91 in the visible light spectrum.

Subsequently, the inversion performance of the NN-V model was evaluated using field measurements from the OC-CCI in situ dataset spanning from 2019 to 2020. The results, depicted in Figure 9, demonstrate the high accuracy of the NN-V model, showcasing excellent performance across different spectral bands. Table 5 presents detailed statistical parameters for each individual spectral band. In the cases of 443 nm, 486 nm, 551 nm, and 671 nm, the inversion accuracy of the NN-V model surpasses that of the near-infrared atmospheric correction model. For instance, the NN-V inversion of the 443 nm remote sensing reflectance demonstrates an APD of 27.90%, compared to NASA's released VIIRS product with an APD of 30.4%. The smaller RMSD further underscores the robustness of the NN-V algorithm. The relatively larger error observed at 412 nm may be attributed to the systematic bias present in the products of this spectral band used in the training data.

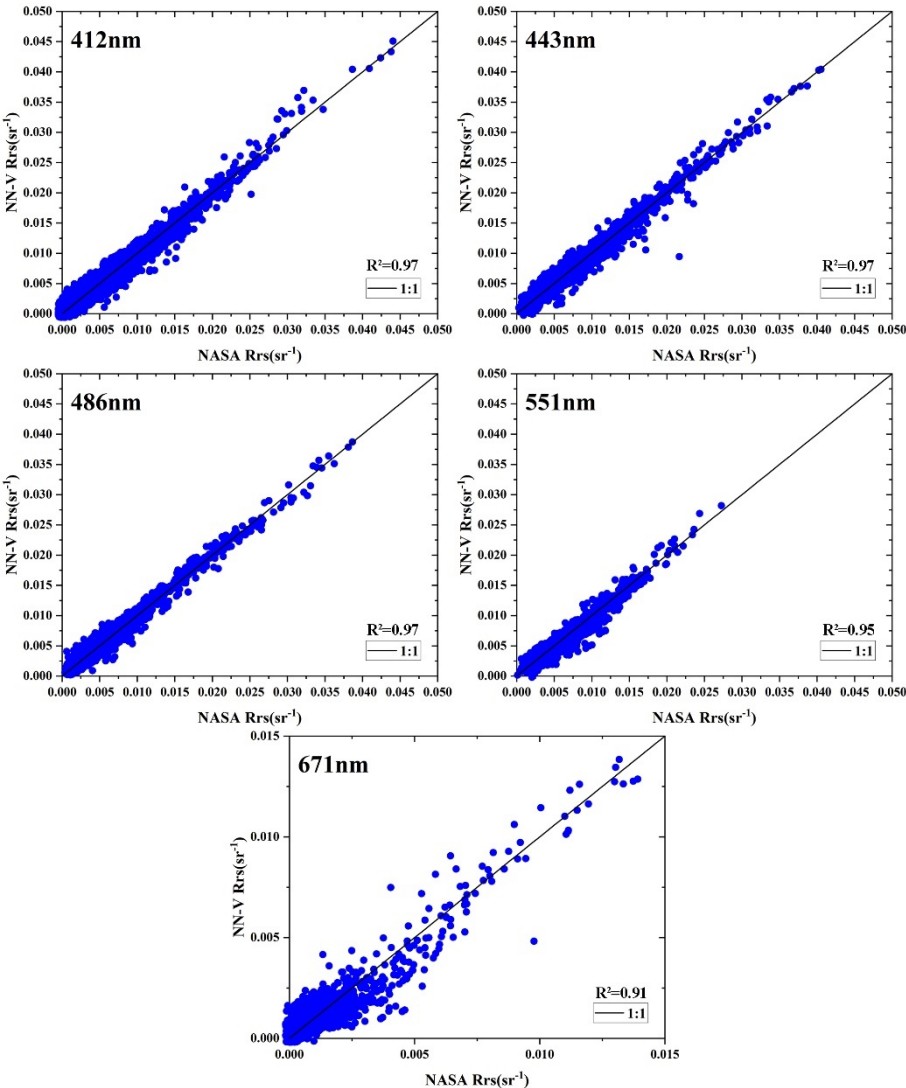

**Figure 8.** Assessment of NN-V model training using model testing dataset (the black line in the graph represents the 1:1 line).

**Table 5.** Summary of validation results for NN-V inversion on VIIRS products.

| Bands | N | $R^2$ | RMSD $(sr^{-1})$ | APD (%) | RPD (%) | Slope |
|-------|-----|-------|------------------|---------|---------|-------|
| 412 nm | 680 | 0.543 | 0.0022 | 49.51 | −21.00 | 0.77 |
| 443 nm | 680 | 0.701 | 0.0019 | 27.90 | 10.64 | 0.84 |
| 486 nm | 680 | 0.831 | 0.0016 | 18.81 | 4.62 | 1.01 |
| 551 nm | 680 | 0.797 | 0.0017 | 18.55 | −1.23 | 0.99 |
| 671 nm | 680 | 0.611 | 0.0006 | 41.91 | 3.65 | 0.69 |

After validation using both field measurements and satellite data, the utility of the NN-V model was confirmed. Subsequently, it was applied to process satellite data with complex observation geometries, as illustrated in Figure 10. The overall statistical parameters are also presented in Figure 10, demonstrating the NN-V model's capability to effectively handle satellite data with large observation geometries, resulting in overall low deviations in the inversion outcomes.

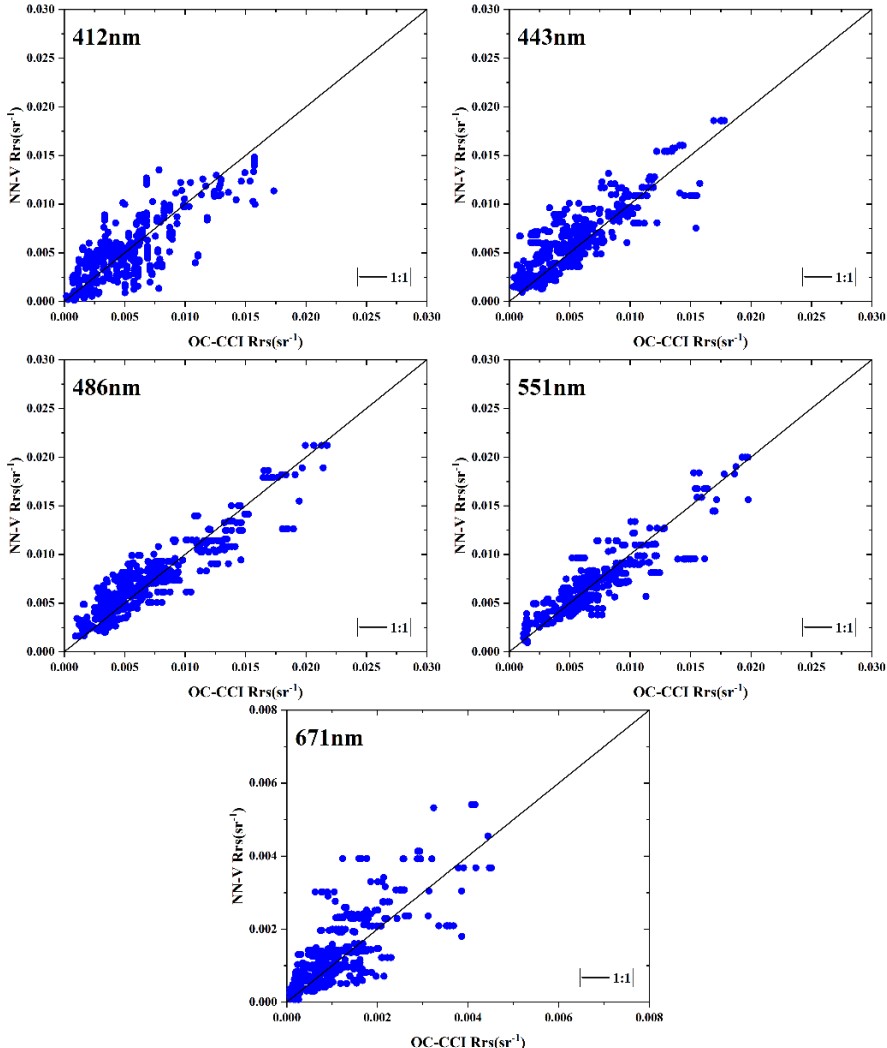

**Figure 9.** Evaluation of NN-V model inversion performance using OC-CCI in situ data (2019–2020). (The black line in the graph represents the 1:1 line).

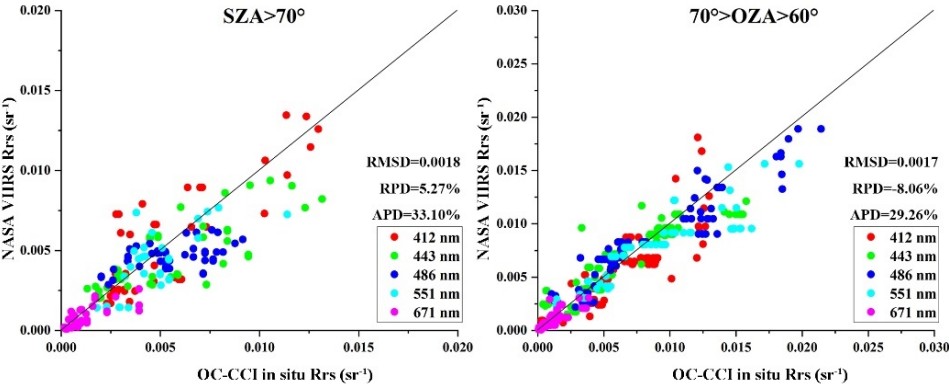

**Figure 10.** Evaluation of NN-V model performance on satellite data with complex observation geometries. (The black line in the graph represents the 1:1 line).

The results obtained through the NN-V model show promising capabilities in handling satellite data with high SZAs exceeding 70 degrees or OZAs greater than 60 degrees but less than 70°. For instance, when the SZA is greater than 70°, the RMSD, APD, and RPD are 0.0020, 48.69%, and −38.59%, respectively, for conventional processing, whereas for

the NN-V model, these values are 0.0018, 33.10%, and −5.27%. This indicates a significant enhancement in its performance. This highlights the robustness of the NN-V model in addressing challenging observation geometries and its potential for accurate retrievals under extreme conditions.

After confirming the effectiveness of the NN-V model in handling satellite data with complex observation geometries, the model was ultimately applied to process a large volume of VIIRS Level 1 data, allowing for the generation of long-term products and providing a means to assess the model's stability. Figures 11 and 12 illustrate the comparison between monthly chlorophyll-a concentration products obtained through OC3V inversion of remotely sensed reflectance using the NN-V model and those released by NASA ($Chl = 10^{(a0 + \sum_{i=1}^{4} ai(log10(max(Rrs(443), Rrs(486))/Rrs(551))))}$). It is evident that during the peak SZAs in the autumn and winter seasons (October and January), the NN-V model significantly improves the availability of valid data, particularly in the winter. Due to the limitations of traditional atmospheric correction models under large SZAs during the winter, the NN-V model produces 20 times more valid data compared to NASA's released products. Additionally, as seen in Figure 12, it can be observed that in regions with valid data coverage, the products released by NASA are generally consistent with the results from NN-V inversion.

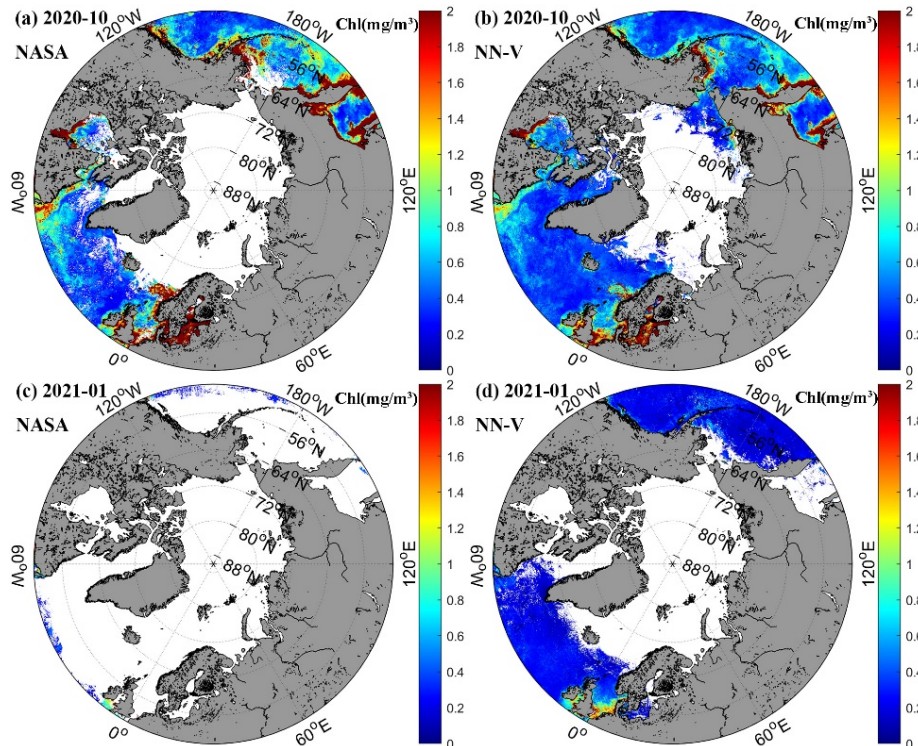

**Figure 11.** Comparison of monthly chlorophyll-a concentration products derived from NN-V inversion and NASA's released data. (**a**) The chlorophyll-a concentration product released by NASA for October 2020. (**b**) same as (**a**) but for the result by NN-V. (**c**) The chlorophyll-a concentration product released by NASA for January 2021. (**d**) same as (**c**) but for the result by NN-V.

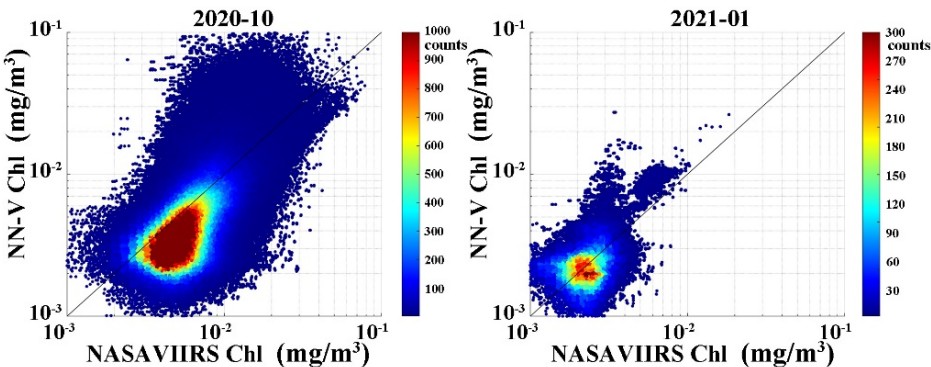

**Figure 12.** Scatter plot comparison of monthly chlorophyll-a concentration products derived from NN-V inversion and NASA's published data (the black line in the graph represents the 1:1 line).

## 4. Discussion

### 4.1. Overall Performance Assessment and Annual Variation

A comparison of our results with previous work revealed a discrepancy in the accuracy of VIIRS products. This difference may stem from significant variations in the number of data used for accuracy assessments. In this study, we utilized a substantial amount of data, approximately 8312 matched pairs, whereas the previous study only had a few tens of matched data. Moreover, our analysis utilized data from both fixed platforms (AERONET-OC) and cruise measurements, in contrast to the limited data (only from fixed platforms) used in earlier studies. Our results demonstrate that the accuracy of the Rrs(486) is higher than that of the previous studies, with an APD of 17.9% (compared to 19% by Barnes et al. (2019) [9] and 20.1% by Hlaing et al. (2013) [4]). However, the Rrs products at other bands have a relatively lower accuracy (for instance, APD(551 nm) is 26.3% compared to 21% reported by Barnes et al. (2019), 9.4% by Hlaing et al. (2013), and 10.6% by Ahmed et al. (2014)) [4,9,26]. The 412 nm and 671 nm bands exhibit higher APD values. The error in the 412 nm band is attributed to the long-standing bias with the VIIRS measurements, and the error in the 671 nm band is influenced by extremely low values of Rrs data. Overall, we employed a larger number of matched data, encompassing information from various data sources. The evaluation results exhibit slight differences in different spectral bands compared to other studies (As shown in Table 6.). This outcome addresses the disparities observed in past satellite product accuracy assessments, which were often attributed to the limited size of matched data in previous studies. Our analysis showed the RPD values of all bands within 20%, which indicates that NASA's multiple calibration efforts (in the years 2012, 2013, 2014, 2018, and 2022) have effectively minimized the systematic biases with the VIIRS products. The observed high APD values for each band measurement emphasize the necessity of improving the atmospheric correction algorithm for retrieving more accurate Rrs values from VIIRS data. As for the annual variations, the APD values were consistently below 17% during the period from 2012 to 2016 (except for 2013) and remained around 20% for the period from 2017 to 2021. Similarly, the $R^2$ value consistently exceeded 0.82 over the years from 2012 to 2016 (except for 2015), and it declined to below 0.78 over the years from 2017 to 2021. This suggests a gradual decline in the accuracy of VIIRS products due to their prolonged operation over the years.

**Table 6.** Summary of APD in VIIRS product validation results.

| Citation | N | 412 nm | 443 nm | 486 nm | 551 nm | 671 nm |
|---|---|---|---|---|---|---|
| Ahmed et al. (2013) [26] | 29 | 39.4% | 20.8% | 12.6% | 10.6% | 18.2% |
| Hlaing et al. (2013) [4] | 16 | 54.8% | 23.4% | 20.1% | 9.4% | 23.9% |
| Barnes et al. (2019) [9] | 55 | 27.0% | 23.0% | 19.0% | 21.0% | 37.0% |

*4.2. Impact of Observation Geometry*

From the analysis of VIIRS Rrs products under varying SZAs, it is evident that the product accuracy is strongly dependent upon solar geometry. To process the data with larger SZAs requires a novel atmospheric correction that will improve the accuracy of Rrs retrievals from VIIRS data. Further, our analysis revealed a similar situation in the VIIRS Rrs product accuracy at larger OZAs. Under high SZAs and OZAs, the accuracy reduction in satellite Rrs retrievals can be attributed to several factors. In such conditions, the solar energy is substantially weak, leading to reduced illumination on the ocean surface. At larger SZAs, the optical pathlength of sunlight is much longer through the atmosphere, meaning that the light intensity decreases and Rayleigh scattering becomes more effective at removing the shorter wavelengths, and dust particles from the lower atmosphere play a crucial role in attenuating solar irradiance and hence determining aerosol scattering and absorption processes. The bidirectional reflectance factor of larger SZAs and OZAs is more amplified. Additionally, there is a pronounced effect of the Earth's curvature on the TOA radiance. As a consequence, upwelling radiance from the water body decreases substantially, which induces strong biases on remote sensing reflectance retrieved by a conventional atmospheric correction algorithm (Li et al. 2019) [27].

*4.3. Usage Scope of Neural Network Atmospheric Correction Model*

This paper, based on the multiple daily observations of the VIIRS satellite in high-latitude maritime areas, establishes a neural network training dataset. Using this dataset, the NN-V atmospheric correction model is developed, specifically tailored for observation conditions with high SZAs. As a result, it exhibits a significant advantage when processing high-latitude maritime areas, such as the Arctic and Antarctic oceans. However, constructing the training dataset involves linking satellite observations from different times within a day, and the presence of highly dynamic nearshore turbid waters introduces anomalies in the training dataset. Consequently, these specific data are excluded during the construction of the neural network training dataset. Therefore, the model is primarily suitable for waters with a relatively low level of turbidity. In the training dataset, the SZA ranges from 0 to 86°, making the model applicable only for processing satellite data within this specific SZA range.

**5. Conclusions**

The evaluation and analysis of the 10-year VIIRS Rrs products have demonstrated a high level of accuracy across various water types, despite the system being operational beyond its design lifetime of 5 to 7 years. The higher performance of VIIRS is achieved because of NASA's on-orbit calibration/characterization activities and regular lunar observations related to wavelength-dependent gain degradation in different bands. However, our study evaluating the long-term VIIRS Rrs products has shown a declining trend in product accuracy. For example, the product accuracy in specific spectral bands, such as the 412 nm and 671 nm bands, is considerably low, with a long-term bias in the 412 nm band. Moreover, there is a declining trend in the annual accuracy of VIIRS Rrs products, particularly in coastal or eutrophic waters, where the spread of the matchup pairs has been more pronounced in recent years. Our analysis indicated that, in observation environments with high SZAs (greater than 70°), the accuracy of VIIRS Rrs products has declined by nearly 50% compared to typical SZA observation conditions. In response to the challenge of declining accuracy in VIIRS products under large observation geometries, we developed the neural network atmospheric correction model (NN-V). This model, constructed based on carefully curated VIIRS products, exhibits exceptional performance when handling VIIRS data in conditions of extensive observation geometries. Particularly during the winter season in high-latitude marine regions, the NN-V model demonstrates a remarkable enhancement in ocean color product coverage, achieving an increase of nearly 20 times compared to traditional methods. In conclusion, the VIIRS sensor has proven capable of working as a successor to MODIS and providing crucial earth science data products beyond

its design lifetime. The high product accuracy in most spectral bands shows the success of NASA's calibration efforts. Nevertheless, our study emphasizes the importance of continuous monitoring, further efforts on improving the atmospheric correction algorithms, and careful consideration of different environmental and geometry conditions (such as SZAs and water types) to ensure the long-term data quality and integrity of VIIRS products.

**Author Contributions:** Conceptualization, H.L. and X.H.; Data curation, H.L. and X.H.; Funding Acquisition, X.H.; Methodology, Y.B.; Manuscript Polishing, P.S.; Software, D.W. and T.L.; Resources, F.G. All authors have read and agreed to the published version of the manuscript.

**Funding:** This research was funded by the National Natural Science Foundation of China (Grants #42176177, #42206183, #41825014, #U22B2012, and #42176182), the Zhejiang Provincial Natural Science Foundation of China (Grant #LDT23D06021D06), the "Pioneer" R&D Program of Zhejiang (2023C03011), and supported by the Science Foundation of Donghai Laboratory (DH-2023QH0002).

**Data Availability Statement:** The model established in this paper and the running programs are stored in the Marine Data Archive under the path Aphia—Public/Antarctic Ocean Color Data/, with the filenames 'main.m' and 'viirsnet.mat'.

**Acknowledgments:** We are grateful to the authors of the cited article (Valente et al. [11–13]) for providing their global bio-optical in situ data, and to the NASA team for granting access to the VIIRS data. These valuable contributions have greatly enhanced our research work on the VIIRS performance assessments. We thank the satellite ground station, satellite data processing and sharing center, and marine satellite data online analysis platform (SatCO2) of SOED/SIO/MNR for assistance with data collection and processing. We also thanks three anonymous reviewers for their constructive comments which help us improve the manuscript largely.

**Conflicts of Interest:** The authors declare no conflict of interest.

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
