# Peer review of "Assessing and Improving the Accuracy of Visible Infrared Imaging Radiometer Suite Ocean Color Products in Environments with High Solar Zenith Angles"

_remotesensing, doi:10.3390/rs16020339_

Round 1

Reviewer 1 Report

Comments and Suggestions for Authors

This paper looks at the quality of reflectance data from the VIIRS instrument by analysing matchup-s between satellite observations and a large dataset of in situ measurements. The authors focus on any change in quality over time and with viewing angle, and suggest a machine-learning approach to improve satellite estimates for large observation zenith angles. The results are potentially interesting and useful but I have questions about some of the analysis and presentation: I recommend a thorough review and revision addressing the following points.

Introduction, final paragraph: Barnes and Hu (2016) https://doi.org/10.1016/j.rse.2015.12.048 is relevant to the discussion of viewing angle but not included. 

Line 147: OC-CCI data – I suggest specifying that this is in situ data from Valente et al. (2020) as there is also an extensive set of OC-CCI satellite data.

Line 155: it’s not clear from the global map that much of the data comes from Case 2 waters. Please could you give a bit more information, e.g. quantify the number that are Case 1 and Case 2 and/or show the Case 2 locations in a different colour?

Line 156: missing “from”, i.e. come from Case-2 waters.

Line 204: Barnes et al. (2015) is not in the list of references.

Line 225: The equation refers to RMSE but RMSD is used in the text. (My preference is for RMSD as is acknowledges that neither the in situ nor the satellite data are equal to the actual value.)

Line 228: Please explain what observed and retrieved values mean in this context – I assume observed is the OC-CCI in situ and retrieved is the VIIRS level 2 satellite.

Section 3.1: the K-Means analysis is not described in the methods. Please give some details of how this was done. Does Figure 2 show in situ data only?

Figure 3 (and other figures): The caption and labelling are not very clear here. Please make clear that “NASA” refers to the VIIRS satellite data and “OC-CCI” is in situ, indicate what is plotted, including the units, and explain the colour scale. It is very helpful if figures can be understood without referring to the text.

Section 3.3: The analysis here does not convince me that there is deterioration over time. It’s not obvious from visual inspection of Figures 4 and 5, or from the data in Table 2. As noted, the different numbers of points in each year make direct comparison difficult, but picking out two years (2014-2020) is not enough to show a trend - couldn’t it just be a feature of these two years, e.g.  more of the data points for 2020 might be in costal locations? I suggest looking for an analysis method that does not involve breaking down the data by years. One possibility would be to plot the absolute difference against time and look for a trend, but an expert in statistics would probably be able to suggest a better method.

Figure 4 and 5: I don’t understand why this is separated into two figures. It would be easier to compare all the years if they were presented together, even if the individual sub-plots were smaller. My comments about captions and labelling of Figure 3 also apply here.

Line 265-266 “the inversion accuracy of VIIRS Rrs products is higher in clean water bodies than in coastal waters”: Is 0.01 sr -1 considered to be the threshold for this? – it’s not quite clear from the text.

Line 269-270: “The accuracy remains stable in clear waters but deteriorates in water bodies with Rrs(486) greater than 0.01 sr-1”. You could consider extending Table 2 to show statistics for reflectances above and below 0.01 sr -1 separately. This would strengthen your argument that there is deterioration for coastal waters – I am not convinced of this just from looking at the figure.

Table 2: 2021 has so few points that its statistics are unlikely to be reliable – I suggest omitting it.

Section 3.4: The suggested relationship, that errors increase for higher SZA, is not clear to me from Figure 6 and Table 3. There is a large scatter for the smallest SZA set, and relatively few high reflectance data points for moderate SZA so the scatter cannot be assessed. I don’t see any clear trend in the statistical metrics (note that metrices is a mis-spelling). Similarly, I do not agree that “Rrs product accuracy noticeably decreases with increasing OZAs” from the evidence presented here (Figure 7 and Table 4).

Lines 296 and 298: please give some information about references 19, 20, 21 and 22 – what are they  about and why are they included?

Section 4.1: I’m confused by this section. What is the purpose of comparison to other studies, and what is the conclusion? The last few sentences are about trend over time, which is not addressed by the previous studies, I think?

Table 5: Please ensure that the table caption specifies what metric is shown.

Section 4.2: some of the material in the first paragraph might be moved to the results section, as it is summarising the results rather than discussing them.

Line 353-4 “Rayleigh becomes more effective at removing the longer (in addition to shorter) wavelengths”. Rayleigh should be Rayleigh scattering. This removes shorter but not longer wavelengths, when viewed along the line of sight (shorter wavelengths are scattered more than longer ones).

Section 4.3: Much of this is results rather than discussion. I suggest moving the presentation of results to section 3, or creating a new section for the ANN work. Please also give more details of how the results presented relate to the training and test datasets described in section 2.3. For example, what is the “NASA VIIRS Level 2 satellite data” shown in Figure 8?

The NN-V method seems promising, but I’d like to see more discussion of the kinds of conditions in which it could be useful. Large solar zenith angles typically mean winter months, as shown in Figure 11, but in these seasons there is typically little phytoplankton activity and chlorophyll values are low. Increasing chlorophyll coverage over areas that are effectively zero does not seem very useful – can you demonstrate that the method can pick up winter blooms that would otherwise be undetected?

Comments on the Quality of English Language

The English is generally understandable, but there are some errors and missing words which detract from readability. 

Reviewer 2 Report

Comments and Suggestions for Authors

This manuscript conducts a comprehensive and long-term accuracy assessment of VIIRS remote sensing reflectance products. They introduce a neural network-based atmospheric correction (AC) scheme designed to handle VIIRS data under extreme observational conditions. The emphasis of their work lies in evaluating the AC applicable for high solar zenith angles. Overall, the approach proves effective, yielding reasonable outputs. The topic aligns well with the scope of the journal, and it is recommended for acceptance with minor revisions. Specific comments for modifications are as follows:

Point 1: Page 2, Line 75. The term "sensor" should be removed.

Point 2: Page 2, Line 91. "RMSE" should be modified to "RMSD." This change should also be applied to Page 5, Line 225.

Point 3: Page 4, Line 176. In section 2.3, the "Neural network atmospheric correction model," it is recommended to provide references and details on the specific structure of the established NN-V model, such as input and output parameters.

Point 4: Page 9, Line 285. In Table 2, it is suggested to supplement the line plot to allow readers a more intuitive observation of the annual variations in VIIRS product accuracy.

Point 5: Page 13, Line 381. It is recommended to display the corresponding correlation coefficients in Figure 8.

Point 6: Page 16, Line 419. Figure 11 illustrates the inversion results of the NASA-released chlorophyll product and the NN-V model established in the article. It is suggested to add a scatter plot to show whether the two products agree well when the solar zenith angle is not too large.

Reviewer 3 Report

Comments and Suggestions for Authors

Overall this is a solid manuscript that provides very useful information on the multi-year performance of VIIRS, and introduces what appears to be a very successful neural net atmospheric correction scheme. I would be glad to see this published after the authors clean up some structural issues, and expand a bit on the atmospheric correction. In particular the results of the NN-V correction are presented, but there’s no code or description so as written, nobody could reproduce the results, even though it’s a main point of the manuscript.

Specific comments:

The definition of acronyms is very uneven. For example MODIS is not defined (line 38) VIIRS is defined on line 177 after multiple uses of VIIRS before that (and after defining in the abstract), RMSD, APD, and RPD are used before being defined, etc. In general the acronym should be defined on first use, and only needs to be defined once. 

Line 176: rewrite. Suggest something like “come from Case-2 waters while a small portion come from open oceanic waters (Case-1).”

Line 191: reference missing

Line 205: Melin is referenced [14] but not Barnes [9]. In general should check all the references

Line 240: K-means clustering requires pre-assigning the number of clusters. Was this done using a statistical approach (such as the shadow method), arbitrarily, or by using the same clusters as a previous publication? 

Figure 2: should specify that the heavy black line is the mean (or median, or whatever)

Table 1 and elsewhere: while APD and RPD are useful, since many of the plots are property-property plots, it would be useful to include the slope, since that’s another metric for relative bias of the full dataset. 

Section 3.3: be more specific in your terms. Accuracy and precision seem to be used interchangeably, but precision is probably more usefully described by the RMSD while accuracy is captured by APD. 

Table 2: you should really be using the adjusted R^2 value given that your n is quite variable from year to year and band to band. The large n values are almost certainly showing inflated R^2 values simply because of the number of samples. 

Line 277-278: that’s a qualitative statement and you have the data to be more specific. I plotted your R^2 values by year and fit a linear regression, the slope has a significant negative value, which is more useful than saying that it appears to be degrading (but note it also looks like a step function, with the initial years relatively stable and then a fairly dramatic decline)

Figures 6-7: why not include the 1:1 line? Would be easier to interpret the data. 

Line 321: those are incorrect terms. 412 is visible, as is 671.

Line 333: again, looks more like a step function rather than a gradual decline.

Line 347: strong

Section 4.3: all of this should really be in the results, not the discussion. You can keep the discussion focused on what is improved, etc. 

Figure 10: RPD is written incorrectly in both panels

Figure 11: what chlorophyll algorithm is being used? It appears that NN-V fills in more data (which is discussed) but some of the values are also different for the same regions. I assume there are probably validation data for chla in the datasets used, and it would be helpful, even if there’s no validation comparison, to show a plot of the NASA vs. NN-V chla distributions to see how much the atmospheric correction is changing the derived biomass values. On line 442 the expanded coverage is highlighted, but there’s no information as to whether the chla values are reasonable compared to validation data. One could get similar improved coverage by using (for example) DINEOF, and there should be some discussion about how believable NN-V is If it’s to be used operationally. 

Comments on the Quality of English Language

Overall very well written but could use some light copy-editing.

Round 2

Reviewer 1 Report

Comments and Suggestions for Authors

My thanks to the authors for their efforts to thoroughly address the points I raised. Most have been fully covered, but there are still some minor corrections and edits needed, which I have listed below. I’d like to highlight two larger points:

(1)    the values of R2adj in Table 2 seem to be inconsistent with the formula (equation 2). For example, using the 2012 values, R2=0.84 and N=1388, so the formula gives R2adj=0.839.  Given the large sizes of N, the adjusted value is going to be very similar to the basic R2, so I’m not sure that it’s useful here.

(2)    I like the new Figure 12, but I think it’s important to discuss the obvious bias in the left hand panel, which shows that results from NN-V are generally lower than NASA VIIRS.

I remain unconvinced about the evidence for a decline an accuracy of Rrs(486) over time, as presented in Figure 4 and Table 2, but the authors have made the changes that I requested and I would be willing to accept publication, following corrections, if other reviewers support it.

Line 112-115: The year is missing in the Barnes and Hu citation, and reference 10 is the wrong paper  - it should be:

Barnes, B. B. and Hu, C.; Dependence of satellite ocean color data products on viewing angles: A comparison between SeaWiFS, MODIS, and VIIRS. Remote Sensing of Environment. 2016, 175, 120-129. https://doi.org/10.1016/j.rse.2015.12.048.

Figure 1: Thank you for distinguishing the Case-1 and Case-2 waters, this is helpful. But is the labelling correct? Many open ocean points are show as blue (Case-2), which the opposite of what I would expect. I’m also surprised to see all Mediterranean points the same – I’d expect to see a mix of Case-1 and Case-2 sites. Perhaps it would be more useful to use the four categories identified in section 3.1.

Lines 156 and 177:  The wavelengths quoted for the OC-CCI in situ data and the VIIRS data are the same, but line 176 says they are slightly different. Please could you clarify this.

Section 2.3 Thank you for adding information about the K-means clustering analysis. Please could you specify what method or analysis package you used.

Figure 2: Please label the axes.

Line 368: I suggest: “there is a significant decline in product accuracy, which is consistent with the findings of previous studies [24, 25]”.

Line 428: I’m confused about why results for SZA between 30° and 40° are given, when the rest of the paragraph is about large SZA or OZA.

Figure 12: the caption says this is a comparison of monthly chlorophyll-a concentration, but the axes say Rrs. Which is correct? Similarly, line 437 says that Figure 11 and Figure 12 illustrate the comparison of Rrs, but Figure 11, and maybe Figure 12, shows chlorophyll.

Line 486-7 “the R2 value consistently exceeded 0.82 over the years from 2012 to 2016 (except for 2015), and it declined to below 0.78 over the years from 2017 to 2021.” This is not true for the adjusted R2 now given in Table 2.

Line 517: there is no entry 27 in the list of references.

Comments on the Quality of English Language

The quality of English is good, but some minor corrections are needed.  

Reviewer 2 Report

Comments and Suggestions for Authors

All of my suggestions have been accepted by the authors, the revised manuscript can be published directly.

Reviewer 3 Report

Comments and Suggestions for Authors

Thank you for carefully considering my recommendations from the first review. I believe you have addressed all of my comments, and this is a much improved version. I recommend publication, but note that in Figure 2, you swapped case-1 and case-2 (case-2 appear to be red symbols, the legend says blue, and vice versa). 
